# Paper-Based Electrodes Decorated with Silver and Zinc Oxide Nanocomposite for Electro-Chemical Sensing of Methamphetamine

**DOI:** 10.3390/s23125519

**Published:** 2023-06-12

**Authors:** Nigar Anzar, Shariq Suleman, Husnara Bano, Suhel Parvez, Manika Khanuja, Roberto Pilloton, Jagriti Narang

**Affiliations:** 1Department of Biotechnology, School of Chemical and Life Science, Jamia Hamdard University, New Delhi 110062, India; 2Department of Biochemistry, School of Chemical and Life Science, Jamia Hamdard University, New Delhi 110062, India; 3Department of Toxicology, School of Chemical and Life science, Jamia Hamdard University, New Delhi 110062, India; 4Centre for Nanoscience and Nanotechnology, Jamia Millia Islamia, New Delhi 110025, India; 5Institute of Crystallography, National Research Council (CNR-IC), 00015 Rome, Italy

**Keywords:** methamphetamine, detection, recreational drug, electrochemical, sensors

## Abstract

We present the development of an electrochemical paper-based analytical device (ePAD) for the detection of methamphetamine. Methamphetamine is a stimulant that young people use as an addictive narcotic, and it must be detected quickly since it may be hazardous. The suggested ePAD has the advantages of being simple, affordable, and recyclable. This ePAD was developed by immobilizing a methamphetamine-binding aptamer onto Ag-ZnO nanocomposite electrodes. The Ag-ZnO nanocomposites were synthesized via a chemical method and were further characterized via scanning electron microscopy, Fourier transform infrared spectroscopy, and UV-vis spectrometry in terms of their size, shape, and colloidal activity. The developed sensor showed a limit of detection of about 0.1 μg/mL, with an optimum response time of about 25 s, and its extensive linear range was between 0.01 and 6 μg/mL. The application of the sensor was recognized by spiking different beverages with methamphetamine. The developed sensor has a shelf life of about 30 days. This cost-effective and portable platform might prove to be highly successful in forensic diagnostic applications and will benefit those who cannot afford expensive medical tests.

## 1. Introduction

Methamphetamine (N-methyl-1-phenylpropan-2-amine) is a stimulant drug comprised entirely of synthetic chemicals [1]. When taken in large doses, meth, which is more potent than any other naturally occurring stimulant, can result in violence, hallucinations, and psychosis [2]. Meth is typically synthesized as a powder that resembles crystals or as tablets. It can be ingested, smoked, injected, or snorted. There are other street names for it, including crank, ice, speed, and glass, but meth is the most frequent. Illegal drugs have a significant negative influence on the environment, including waste waters that are produced all over the world and may contain illegal substances that come from human consumption and excretion, inappropriate disposal, and waste from industries throughout the manufacturing process. Although illicit substances are normally prevalent in the environment at low quantities of between 1 ng and 1 g, the presence of these chemicals at ambient levels can have undesirable physiological consequences on both humans and animals [3]. Conventional methods from the last decade that are used for the detection of these illicit drugs include immense and time-intense procedures such as GC-MS (gas chromatography–mass spectrometry) [4], HPLC (high-performance liquid chromatography) [5,6], IMS (ion mobility spectrometry) [7], capillary electrophoresis, and immunoassays. Due to some drawbacks, such as their complex natures and time-consuming procedures, there is a vital requirement for an appropriate technique for the sensitive, selective, and rapid detection of illicit drugs [8,9,10].

An electrode serves as the conductor in an electrochemical sensor. Contemporary types of electrochemical sensors are often called chemically modified electrodes (CMEs) as they are created by coating electrodes with chemicals to change their surface behavior [11,12]. Nanoparticles are a suitable choice for altering electrochemical sensors because of their high surface area/volume ratios and remarkable abilities to be modified [13,14]. Composite nanomaterials can provide excellent multiplexing and signal amplification for assessments with increased sensitivity and resolution. We chose a silver and zinc oxide nanocomposite (Ag-ZnO NC) for this experiment since it helps to increase the electrochemical signal of this platform [15,16]. Due to their superior surface-enhanced Raman scattering (SERS) ability, biocompatibility, higher level of conductivity, better electrochemical indicators, and catalytic activity, Ag-ZnO NCs have attracted significant research attention in biomedical applications [17,18,19,20,21].

For the detection of minor molecule targets, such as metabolites, pharmaceuticals, and naturally occurring toxins, electrochemical aptamer-based sensor technology is frequently utilized. This technology functions by forcing DNA and RNA aptamers to fold in response to a target. The sensing component in this technique is a modified redox reporter, a structure-switching aptamer. Aptamers are employed as novel recognition elements because they have benefits over antibodies, including good stability, low dimensions, and strong affinities. They form secondary structures which allow them to recognize and attach to tiny targets such as meth. Additionally, they are simple to modify and easy to synthesize in vitro [22,23]. The rate of electron transport between the aptamer and the surface of the electrode are modified by the asymptotic connection between the concentration range and the aptamer’s target-induced configuration change [24]. Meth is an electrochemically active compound that can undergo the transfer of electrons to an electrode surface at higher positive potentials [25]. Electroanalytical approaches have been shown to be effective for identifying a range of electroactive compounds. They are simple, affordable, and only require a brief amount of analytical time as no time-consuming derivatization or extraction activities are necessary. Some electroanalytical methods for determining meth have been published, although the majority of them are based upon principles such as immunosensing and electrochemiluminescence [26,27,28,29].

There are many issues with the traditional available electrodes such as glassy carbon electrodes, including that they are bulky, expensive, and non-disposable [30]. Therefore, in order to avoid the issues with traditional electrodes, paper-based electrochemical analytical devices (ePADs) are employed. ePADs usually consist of a three-electrode setup integrated onto a paper substrate. Fabrication techniques such as stencil-printing, sputtering, screen-printing, or inkjet-printing are often employed to spread a conductive ink on the paper substrate [31]. In this study, we proposed a paper based electrochemical analytical device (ePAD) for the detection of meth. Ag-ZnO NCs were employed for the modification of the working electrode of the developed ePAD. Furthermore, a meth-binding aptamer was incapacitated on the surface-modified working electrode (with a nanocomposite attached), and meth was then detected using an electrolytic buffer. The results were then verified using electrochemical techniques, i.e., cyclic voltammetry (CV) and linear sweep voltammetry (LSV), which confirmed each attachment. Additionally, we presented the applicability of this proposed sensor in beverages. This proposed biosensor is a promising option for the sensitive and accurate detection of meth.

## 2. Material and Method

### 2.1. Chemicals/Reagents Required

All the chemicals used in this research were of AR Grade. For the construction of the three-electrode system, carbon conductive ink was purchased from Snab graphix Pvt Ltd. Bangalore, India. For the synthesis of silver nanoparticles and zinc oxide nanorods, silver nitrate, sodium borohydride, hydrochloric Acid (HCL), sulfuric acid, and zinc nitrate were purchased from Loba chemie, India. Methylene blue and potassium chloride were purchased from Sigma Aldrich, Berlington, MA, US. Methamphetamine was acquired from Mtor life science, New Delhi, India.

### 2.2. Meth Binding Aptamer Sequence

ACG GTT GCA AGT GGG ACT CTG GTA GGC TGG GTT AAT TTG G [32].

A meth-specific binding aptamer sequence was used as a recognition element and was obtained from Amplicon Biotech, India.

### 2.3. Apparatus/Instrument Used

A Metrohm Dropsens (Stat-I 400 s) was employed for the electrochemical measurements (CV & LSV), The shape of the surface of the synthesized material, i.e., the silver–zinc oxide nanocomposite, was determined via field scanning electron microscope (FESEM) technology using Quanta 3D FEG. UV-vis spectroscopy was performed using a UV vis spectrometer, and Fourier-transform infrared spectroscopy (FTIR) was carried out employing a Bruker Tensor 37.

Spiked Beverages:

An alcoholic drink was obtained for spike testing.

### 2.4. Preparation of Standard Solutions

The meth-binding aptamer was prepared by mixing 221 µL of sterile distilled water in a main vial in order to reach 100 µM. For further dilution, 10 µL of aptamer was added to 190 µL of sterile distilled water. The final diluted form was used for the experiment.

For the preparation of the meth concentration, different concentrations, i.e., 0.01—6 µg/mL, were prepared by adding meth to distilled water. An alcoholic drink was also diluted in a 1:1 ratio in distilled water in order to obtain the results.

### 2.5. Synthesis of Ag-ZnO Nanocomposite

For the synthesis of the nanocomposite, silver nanoparticles were prepared via a chemical reduction method. First, 30 mL of 2 mM NaBH_4_ (freshly prepared and ice-cold) was added dropwise to 10 mL of 1 mM AgNO_3_ while continuously stirring. The mixture generated a brilliant yellow color, which indicated the fabrication of silver nanoparticles [33].

After the synthesis of the silver nanoparticles, zinc oxide nanorods were prepared. A chemical process was used to synthesize zinc oxide nanorods, beginning with a 0.2 M solution of zinc chloride in distilled water. Ammonium hydroxide was then added dropwise at room temperature while being continuously stirred to produce zinc hydroxide precipitates. The resulting precipitates were filtered to separate them from the remaining liquid, and at a temperature of 120 °C, they were then dried into a powder form. After crushing the powder, nanorods were synthesized [34,35]. Both the silver nanoparticles and the zinc oxide nanorods were mixed in a 1:2 ratio and were further sonicated for about 1 h. Characterization was carried out in order to confirm the successful synthesis of the nanocomposite [36].

### 2.6. Fabrication of Paper-Based Three-Electrode System

By utilizing a silk screen and conducting carbon ink on paper, paper-based electrodes were constructed. A silk screen with the printed electrodes already attached was used as a stencil to prepare the electrodes (a single electrode had a length of 3.5 cm and a breadth of 2.5 cm). The three-electrode system, i.e., a counter electrode, a working electrode, and a reference electrode drop-casted with silver paste (Ag/AgCl) comprised the printed electrode (As shown in Figure 1). In this study, the sensor is inexpensive, reusable, requires only a small sample volume, and is easy to fabricate due to the use of paper as a sensing surface. Additionally, conductive carbon ink also provides beneficial properties, including its low cost, simple preparation, and speedy manufacturing.

### 2.7. Immobilization of Ag/ZnO Nanocomposite and the Deposition of an Aptamer on a Paper-Based Sensor

First, 30 µL of the Ag/ZnO nanocomposite was placed on a working electrode of a paper-based sensor. The sensor was then dried using a hot plate set at approximately 60 °C. The next step involved the immobilization of the aptamer (20 µL) on the working electrode, which was adorned with the Ag/ZnO nanocomposite and dried for an hour at room temperature [36,37,38]. The detection of methamphetamine was made possible through this sensor modification. By using electrochemical techniques via CV and LSV, different concentrations of meth (0.01 to 6 µg/mL) were dropped over this sensor in combination with a buffer, and the results were detected.

### 2.8. Stages of Electrochemical Detection

In order to confirm the proper deposition of the nanomaterial and aptamer on the sensor, different stages of electrochemical analysis were carried out. For this, CV and LSV values of the uncoated, bare electrodes were assessed. Following the overnight drying of the Ag-ZnO NCs on the paper-based sensor, both voltammetry tests were performed as usual. The aptamer was then applied to the paper-based sensor that contained the dried Ag–ZnO NCs, and CV/LSV measurements were then recorded. The last step involved the deposition of the target molecule—meth—onto the electrodes containing both the aptamer and the Ag/ZnO NCs. A supporting electrolytic buffer was then used to conduct the CV/LSV, and the results were analyzed.

### 2.9. Optimization of Various Parameters and the Procedure for a Real Sample Analysis in Spiked Beverages, Repeatability, and Stability Investigation

Various meth concentrations, ranging from 0.01 µg/mL to 6 µg/mL, were immobilized across various electrodes (which had already been modified with nanocomposites and aptamers) and were then held for drying. After that, the meth determination response was observed at each concentration via CV and LSV. The aptamer-immobilized electrodes were utilized to detect meth at various temperatures (5 to 40 °C) and times (5 to 35 s), and the best cyclic response was observed. The target was determined using the electrodes that had been aptamer-immobilized. To corroborate the results, an electrochemical study was carried out after adding the target meth and buffer to the electrodes. By incorporating a predetermined amount of meth into a spiked beverage (alcohol), the sensor’s capacity to function in a real sample was also evaluated. Meth concentrations were repeatedly measured, proving the suggested biosensor’s repeatability and its stability for at least a month.

### 2.10. Sensing Strategy and Acquiring Signals

Meth is an electrochemically active compound. The chemical structure of meth receives an indication that the secondary amine in the aliphatic part of the molecule is the most likely electro-oxidizable group. This electroactive drug involves the oxidation of primary and/secondary amino groups and the oxidation of the aromatic nucleus, respectively (Figure 2) [39,40]. Due to the introduction of the aptamer and meth onto the ePAD, the electrochemical process is accelerated, and the current response is amplified. The rates of electron transport between the aptamer and the surface of the electrode are modified by the asymptotic connection between the concentration range and the aptamer’s target-induced configuration change. As the meth concentration increases, the current also increases on the ePAD sensing surface. The high surface area and rapid electron transfer kinetics of the Ag-ZnO NCs are also the fundamental drivers of the enhanced current in the circular region of the developed ePAD which is integrated with them. On the modified ePAD, which is merged with the Ag-ZnO nanocomposite and aptamer, meth is oxidized when voltage is applied to the sensor surface [41].

## 3. Results and Discussion

### 3.1. Characterization of Silver and Zinc Oxide Nanocomposite

Using the characterization techniques such as FESEM, FTIR, and UV-vis spectrometry, the effectively synthesized silver and zinc oxide nanocomposites were verified.

As shown in Figure 3a, the morphology of the produced Ag–ZnO NC was examined using FESEM. On a scale of 500 nm, the FESEM image showed the production of ZnO nanorods. A homogeneous distribution of silver NPs 50–100 nm in diameter was produced. The FESEM further showed the development of spherical, slightly agglomerated silver NPs on the surfaces of the zinc oxide nanorods.

Figure 3b demonstrates the Ag–ZnO NCs’ FTIR spectroscopy. The existence of the Ag-O bond and the Zn-O stretching vibrations are shown by the FTIR spectrum peaks at around 618 cm^−1^ and 1654 cm^−1^. The creation of Zn’s tetrahedral coordination at 1012 cm^−1^ is indicated by the tiny peak at 892 cm^−1^, which may be the result of the C = C aromatic stretch. Hydroxyl (O-H) group vibrations were attributed to the bands in the range of 3050 to 3800 cm^−1^. The C = O stretching mode may be the cause of the peaks that were stimulated at 1485 cm^−1^ and 1504 cm^−1^ [42,43].

As can be seen in Figure 3c, the Ag–ZnO nanocomposite was further studied using UV-vis absorption spectroscopy in the 200–800 nm wavelength (λ) region (c). It is assumed that ZnO was present since its distinctive peak at 374 nm was noticed. Due to surface plasmon vibrations, the silver NPs showed a peak at 450 nm. As the Ag–ZnO NCs were successfully synthesized, a band of ZnO at 374 nm with a shoulder peak of silver at 450 nm was observed [44,45].

### 3.2. Electrochemical Properties of Meth/Apt/Ag-ZnO NC at Different Stages

By employing electrochemical techniques such as cyclic voltammetry and linear sweep voltammetry, the electrochemical characterization of the modified meth/aptamer/Ag–ZnO NC paper electrodes was carried out. Current *I* response discrimination at various electrode phases was confirmed by both techniques, as shown in Figure 4a,b. In CV, the bare electrodes had a low current response peak, which can be attributed to the slower electron transfer kinetics. Due to the rapid transfer of electrons kinetics offered by the Ag–ZnO NCs, there was a noticeable increase in current sensitivity after the deposition of the nanocomposites onto the working surface. Due to the aptamer’s (biological recognition element) non-conductive character, the current was drastically decreased when the aptamer was immobilized onto the working surface. Owing to meth’s electroactive properties, the current response increased once the target substance, meth, was introduced.

### 3.3. Effects of Methamphetamine Concentrations on the Aptamer/Ag-ZnO NC Paper-Based Sensor

To illustrate the quantifiable performance of the designed biosensor, several concentrations of the target, methamphetamine, were examined, ranging from 0.01 to 6 µg/mL. The findings indicated that at various concentrations, varied current responses were observed. As the concentration increased, the current also increased, which confirms the sensor’s quantitative functionality. The findings are in agreement with the previously reported sensors. The current response increased as the methamphetamine concentrations increased. Through the use of two electrochemical measures, namely, CV and LSV, the concentration findings were confirmed (Figure 5a,b). An interesting linear association was found between the peak current values and the log of the meth concentration (as shown in Figure 5c,d).

### 3.4. Optimization of Meth/Apt/Ag-ZnO NC Paper-Based Sensor in Terms of Temperature and Time

A crucial stage for the developed sensor’s effective operation is the biosensor’s optimization. Time and temperature have an impact on the functioning of the sensor. The sensor was adjusted in terms of these experimental parameters, including the time and temperature, in order to attain optimal response. The performance of the meth/Apt/Ag-ZnO NC paper-based sensor was investigated thoroughly in a range of temperatures and time conditions. The meth/Apta/Ag-ZnO NC paper-based sensor’s cyclic voltammetry graphs were recorded at various temperatures, varying from 15 °C to 55 °C, at a scan rate of about 50 mV/s^−1^. The best current response was observed at 35 °C. Therefore, the sensor was optimized at 35 °C. Further, the optimization of the sensor was also carried out over different time intervals (5 to 35 s), but the ideal current response time was between 20 and 30 s because at 35 s, it takes longer to display the current response, delaying the sensor’s capacity to detect changes. As a result, it would not be able to detect the meth in as much time (as shown in Figure 6a,b).

### 3.5. Limit of Detection and Test of Accuracy (Recovery)

The limit of detection for the proposed sensor was found to be 0.1 µg/mL by using the formula LOD = 3.3 × standard deviation/slope of calibration curve. The accuracy of the proposed sensor was also evaluated using a recovery test. The other concentration was laced with meth at various concentrations. In order to demonstrate the excellent recovery of the suggested biosensor, 0.01 µg/mL of meth was added to the other meth concentration, which caused the current to be virtually equivalent to 0.1 µg/mL. The same procedure was repeated with other concentration, and the recovery percentages were calculated and found to be 98% and 96%, respectively. The results are shown in Table 1.

### 3.6. Analysis of Stability and Specificity/Reliability (Cross Reactivity) of ePAD

The proposed paper-based sensor’s cross-reactivity performance was evaluated using ketamine (0.01 µg/mL). The samples’ current responses, assessed using cyclic voltammetry, are shown in fig 7a, which demonstrates that the peak flow of the drug ketamine was almost equal to that of the aptamer/Ag-ZnO NC/ePAD, although the current was increased when compared to meth. Additionally, the paper-based sensor was maintained at 4 °C for different lengths of time—1 day, 7 days, 15 days, and 30 days—in order to detect meth (0.01 µg/mL) and test the stability of the sensor. The results clearly show that the built-in sensor generated nearly identical results to the aptamer/Ag-ZnO NC/ePAD and was stable up to the 30th day. The stability graph further shows sufficient repeatability and reproducibility for meth detection (Figure 7b).

### 3.7. Analysis in Spiked Beverage

To test the efficiency of the developed paper-based sensor, we spiked a beverage and examined the sample. In specifically, 0.01 µg/mL of meth was added to a beverage, and peak current value testing (cyclic voltammetry) was carried out on the aptamer/Ag–ZnO NC ePAD surface. The sensor functioned excellently in the spiked beverage, and the result is very comparable to meth. Data were gathered that satisfy the requirements for identification in a real sample. Since the current response was discovered to be virtually identical to the meth ePAD, the constructed sensor was able to detect meth in a drink. (Figure 8a,b).

## 4. Conclusions and Future Perspectives

Electrochemical sensors have drawn significant interest because of their vital role in the early detection of illegal substances in drinks and other fluids. Here, we rationally exploited the exceptionally high charge-transfer efficiency of a Ag/ZnO ternary nanocomposite to develop a detection platform for methamphetamine—a recreational drug. The use of an electrochemical paper analytical device (ePAD) improves the sensor even more because paper is a cheap substrate that can be made in huge quantities with less effort. Paper-based testing offers a point-of-care diagnostic platform that is affordable. Due to their effective manufacturing, use of environmentally acceptable substrates, and ability to minimize waste management, these sorts of sensors are referred to as eco-designed analytical instruments. It is obvious that the suggested sensor requires significantly less time and money than current analytical techniques for meth detection. The detection limit of the reported sensor was found to be 0.1 µg/mL, which is much lower than other previously reported studies. UV-vis spectroscopy, FTIR, and SEM were used to characterize the produced nanocomposites of silver and zinc oxide nanorods. Aptamers were utilized as the recognition element for this study because they are considered unique and highly sensitive instruments for use in quick diagnostic approaches. Additionally, they are easy to manufacture and are selective. Measurements of the analytical response of the biosensor were carried out using cyclic voltammetry and linear sweep voltammetry, both of which were verified with a potentiostat. The sensor is considered to have good sensitivity and selectivity toward meth. Furthermore, the applicability of the sensor was studied using a spiked sample, which revealed that the sensor can be used in forensic labs in the future for drug analysis.

## Figures and Tables

**Figure 1 sensors-23-05519-f001:**
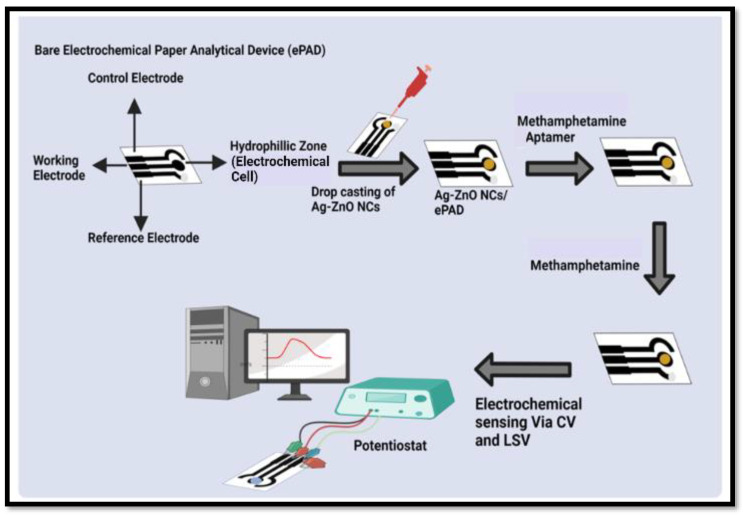
Schematic representation of the fabrication and the operation of the three-electrode system.

**Figure 2 sensors-23-05519-f002:**
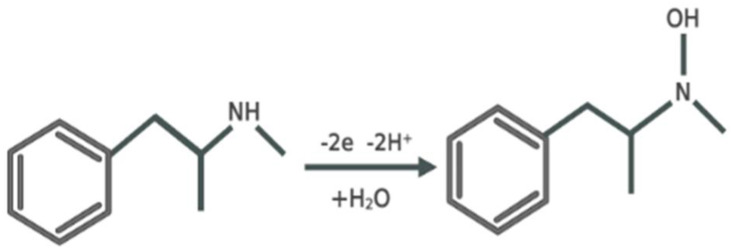
Reaction mechanism for electrochemical oxidation process of meth.

**Figure 3 sensors-23-05519-f003:**
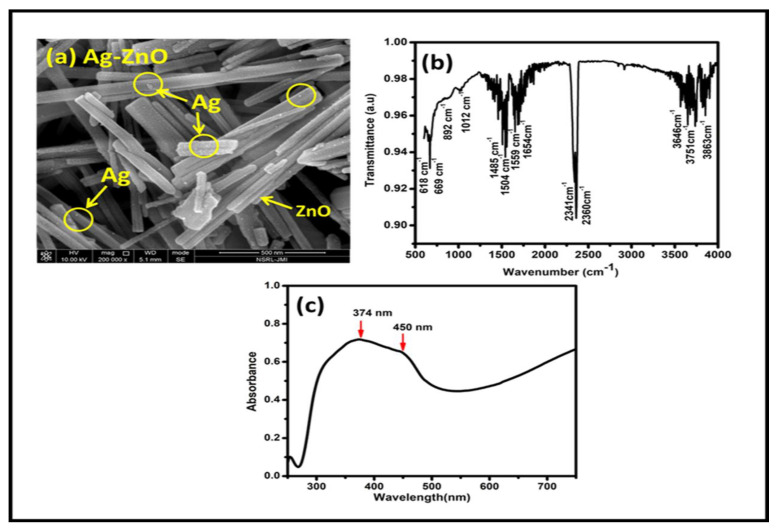
(**a**) FESEM micrograph (**b**) Fourier–transform infrared (FTIR) spectrum, and (**c**) UV−vis absorption spectrum of Ag−ZnO nanocomposite.

**Figure 4 sensors-23-05519-f004:**
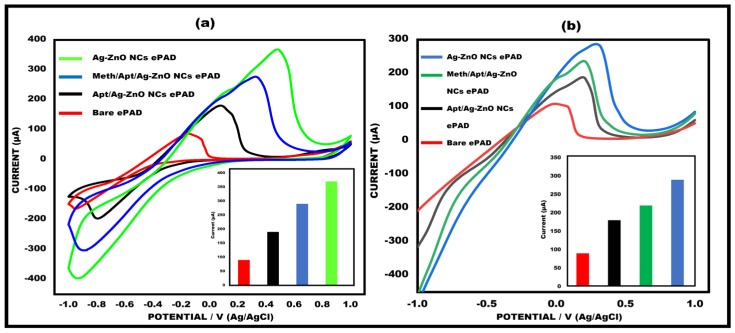
(**a**) Cyclic voltammetry (CV) of bare ePAD, Ag−ZnO NC modified ePAD, apt/Ag−ZnO NC modified ePAD, and target ePAD, i.e., meth/apt/Ag−ZnO NC ePAD at 50 mVs^−1^ in the potential range between −1 V and +1 V with inserted bar graph. (**b**) Linear sweep voltammetry (LSV) in the range between −1 V and +1 V at 50 mVs^−1^ for a bare ePAD, Ag−ZnO NC modified ePAD, apt/Ag−ZnO NC modified ePAD, and target ePAD, i.e., meth/apt/Ag−ZnO NC ePAD with inserted bar graph.

**Figure 5 sensors-23-05519-f005:**
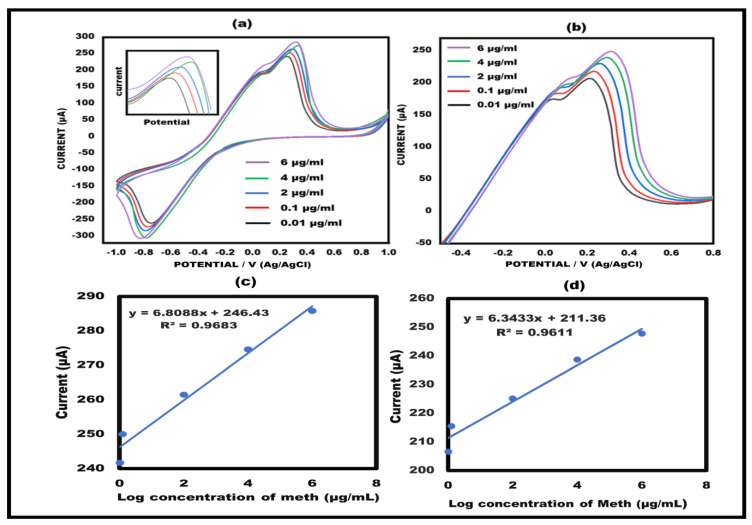
(**a**) Cyclic voltammetry of Meth/Apt/Ag−ZnO NC ePAD at 50 mVs^−1^ in the potential ranging from −1 V to +1 V at different concentration from 0.01 to 6 µg/mL. (**b**) Linear sweep voltammetry of Meth/Apt/Ag−ZnO NCs ePAD at 50 mVs^−1^ in the potential range from −1 V to +1 V at different concentrations ranging from 0.01 to 6 µg/mL. (**c**) Linear curve of the current value and log of the target meth concentration from cyclic voltammetry results. (**d**) Linear curve of the current value and log of the target meth concentration from linear sweep voltammetry response.

**Figure 6 sensors-23-05519-f006:**
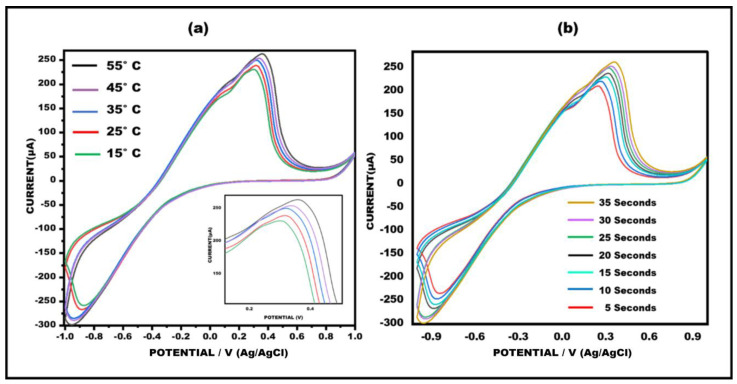
Cyclic voltammetry recorded for the meth/apt/Ag−ZnO NC ePAD for various (**a**) temperatures (15−55 °C) and (**b**) times (5−35 s) at 50 mVs^−1^.

**Figure 7 sensors-23-05519-f007:**
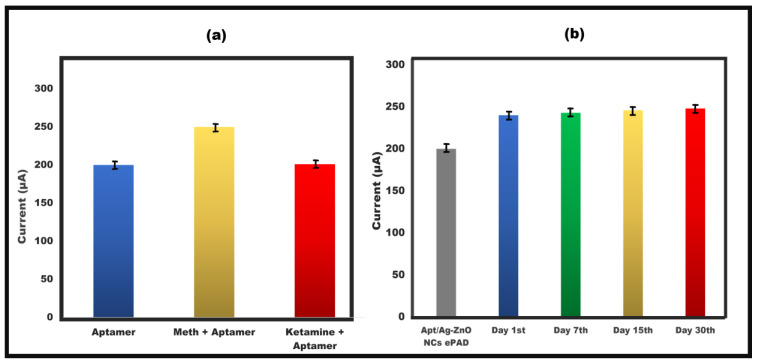
(**a**) Investigation of meth peak current CV value of aptamer/Ag−ZnO NC sensor interaction with the interferent, ketamine, with error bars (n = 5). (**b**) The electrochemical test was used to evaluate the stability of the paper-based sensor’s capacity to detect meth on the first, seventh, fifteenth, and thirtieth days with error bars (n = 5).

**Figure 8 sensors-23-05519-f008:**
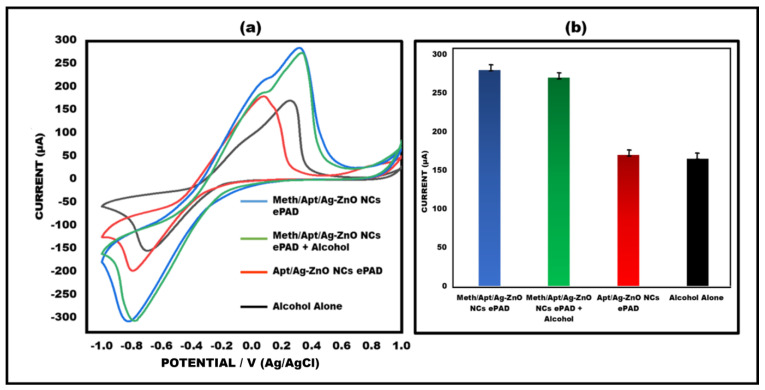
(**a**) Meth in a spiked beverage, detected using the constructed paper−based sensor in a cyclic voltammetry peak current study. (**b**) Error bars are shown in a bar graph comparing the current study of spiked beverages to the other ePADs (n = 5).

**Table 1 sensors-23-05519-t001:** Recovery test of the proposed meth biosensor.

Initial Meth ePAD Concentration (µg/mL)	Concentration of Meth Added (µg/mL)	Final Measured Current (µA)	Expected Current (µA)	Recovery (%)
0.01	0.1	242.5	255.0	98
0.01	2	258.3	265.8	96

## Data Availability

Data can be available on request.

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
