# Peer review of "Paper-Based Electrodes Decorated with Silver and Zinc Oxide Nanocomposite for Electro-Chemical Sensing of Methamphetamine"

_sensors, 2023, doi:10.3390/s23125519_

Round 1

Reviewer 1 Report

The authors reported a work entitled “Paper-Based Electrodes Decorated with Silver and Zinc Oxide Nanocomposite for Electro-Chemical Sensing of Illicit Drug”. The work is systematic and could be considered for publication after minor revisions.

1.     The abstract is too vast, write it precisely to address the objective of this work.

2.     The introduction part needs to be revised to emphasize the importance of ZnO and Ag with recent relevant references in terms of their multifunctional functions with the following references.

Li, Xiaodi, Hualan Zhou, Lehui Wang, Huiwen Wang, Ayiqiaolipani Adili, Jingtao Li, and Jianguo Zhang. "SERS paper sensor based on three-dimensional ZnO@ Ag nanoflowers assembling on polyester fiber membrane for rapid detection of florfenicol residues in chicken." Journal of Food Composition and Analysis 115 (2023): 104911.

Muthusankar, E., Saikh Mohammad Wabaidur, Zeid Abdullah Alothman, Mohd Rafie Johan, Vinoth Kumar Ponnusamy, and D. Ragupathy. "Fabrication of amperometric sensor for glucose detection based on phosphotungstic acid–assisted PDPA/ZnO nanohybrid composite." Ionics 26 (2020): 6341-6349.

Liu, Chunyan, Xiaohui Xu, Changding Wang, Guoyu Qiu, Weichun Ye, Yumin Li, and Degui Wang. "ZnO/Ag nanorods as a prominent SERS substrate contributed by synergistic charge transfer effect for simultaneous detection of oral antidiabetic drugs pioglitazone and phenformin." Sensors and Actuators B: Chemical 307 (2020): 127634.

Chen, Xingang, Lei Zhu, Zhipeng Ma, Meilin Wang, Rui Zhao, Yueyue Zou, and Yijie Fan. "Ag nanoparticles decorated ZnO nanorods as multifunctional SERS substrates for ultrasensitive detection and catalytic degradation of Rhodamine B." Nanomaterials 12, no. 14 (2022): 2394.

Kumar, S. Vignesh, T. Ramya Sri, N. Prakash, and E. Muthusankar. "Preparation and evaluation of silver nanoparticles embedded in Muntingia calabura leaf extract to cure White Piedra." Journal of Pharmaceutical Innovation (2021): 1-12.

3.     2.2. Meth binding aptamer sequence: write a little about this sequence and its relevance in this work.

4.     Apparatus/Instrument Used: Mention the instrumentation for FTIR.

5.     2.5.3. Preparation of Ag-ZnO Nanocomposite: write the detailed procedure with appropriate reference if needed.

6.     2.7. Immobilization of Ag/ZnO nanocomposite and deposition of aptamer on a paper-based sensor. Cite a relevant reference for this section.

7.     Many typo errors are found Ex. Line 216-220 around 618 cm-1 and 1654 cm-1. Here, -1 should be superscript, and at line 219 “bands in the range of 3050 218 to 3800 cm1.” There is no inverse 1. Revise the whole FTIR discussion and fix the typo errors.

8.     Fig.8b makes the x-axis caption “days or no.of days”.

9.     Comment about sensitivity values.

10.  Revise the conclusion and add content about physicochemical and electrochemical results.

-

Author Response

attached file

Reviewer 2 Report

In a present state the manuscript submitted is not to be recommended for publication. 

Listed of the corrections and question:

 Title should be mention methamphetamine  : .... chemical sensing of methamphetamine....

1.     Abstract

i.                 Fisrts six(6) sentences should be release.

ii.                Be carefull with  using capital/small letter.

iii.               Don’t used abbreviation in abstract and should be writing in full sentences.

iv.              Linear range and LOD must be different. Must used formula to calculate LOD.

v.                Rewrite the abstract.

2.     Experiment

i.                 In “Experiment”, ONLY METHOD can be write here.

3.     Results

i.                 Normally used a common figure current vs concentration and don’t used bar cart.

ii.                Why all peak move to right when concentration increase (especially Figure 5b) ?

iii.               Certain figure should insert error bar.

4.     Real samples analysis should be compared (validated) with commercial instrument.

5.     Some of electrochemical performances of sensor (i.e. EIS) must be study such as impedance spectroscopy which from that data such as surface area of sensor, kinetic study etc will strengthen the results.

6.     Check carefully the using of abbreviation in text.

Author Response

attached file

Reviewer 3 Report

This work reports the fabrication and application of an electrochemical paper-based analytical device (ePAD) for the detection of Methamphetamine as a leisure or stimulant drug. This ePAD was developed by immobilizing methamphetamine-binding aptamer on Ag-ZnO nanocomposite-ePAD. The manuscript is not well written and not well formatted. Regarding the analytical method, the manuscript contains a very serious error of not having a calibration plot. Without a calibration plot, the manuscript cannot be accepted. Validation of the analytical method is also quite confusing. Recovery tests must be carried out on the sample. And the accuracy of the ePAD needs to be evaluated from repeatability tests. The use of pulse voltammetric techniques (DPV and SWV) can help in the sensitivity of the method. The manuscript needs much improvement. Therefore, I recommend that a major review be performed before the manuscript can be considered for publication in Sensors.

General comments:

1. Update the terminologies for the electrochemical methods according to new IUPAC recommendations. See the information in Pure and Applied Chemistry, 92 (2020) 641–694. Current is I (in italics) and square wave current is ISW.

2. The authors, like many others, confuse the terms "detection" and "determination". Detection is qualitative by nature, while determination always is quantitative. Qualitative analysis is the detection of the presence of ions or compounds in an unknown sample, for example. The term "determination" refers to quantitative analysis to obtain data on the amount of analyte by weight or by the concentration of an element or a compound in a sample. Therefore, most of the words “detection" in the manuscript should be replaced by the term "determination" (or "quantitation" or "assay") if quantitative assays are involved.

Specific comments:

1. Abstract:

a. The Abstract is very extensive, with excerpts from the Introduction, not the Abstract.

2. Introduction:

a. Lines 59-61. We commonly describe the negative features of chromatographic or atomic absorption spectrometry techniques to laud our (electroanalytical) technique. I believe that all analytical techniques have advantages and disadvantages and that they all have space and function in scientific research. Thus, we can describe the positive characteristics of electrochemical methods, without diminishing the other techniques. This is just an opinion. Authors do not need to answer or comment.

b. Line 94. The authors need to explain in the Introduction what is the definition of an ePAD. To this, add the following sentence: “ePADs usually consist of a three-electrode setup integrated into a paper substrate. Fabrication techniques such as stencil-printing, sputtering, screen-printing or inkjet-printing are often employed to spread a conductive ink on the paper substrate”. Add the reference Microchemical Journal, 179 (2022) 107588 to validate this information.

2. Experimental:

a. Section 2.2 needs to be better explained.

b. Standardize the description of the equipment: model (company, country).

c. It is necessary to create a sample preparation section.

d. The correct one is mL and µL, and not ml and µl. Proofread all text.

e. Inform the geometric area of the ePAD working electrode.

f. Line 168. What was the supporting electrolyte used? And what is the pH value of the electrolyte?

3. Results and discussion:

a. Lines 216, 217, 220. Check the formatting of -1.

b. A bar graph inserted in Figure 4 will enhance the reader's understanding. Please insert a current vs. electrode.

c. Insert the reference electrode used in the x-axis (potential) of the voltammograms. Example: Potential vs. Ag/AgCl / V.

d. Figure 4. The Meth oxidation reaction was hampered by the electrode modifications. This can be seen by Meth's peak potential shift to more positive potential values. What's the explanation? If the nanocomposite is more conducive.

e. A study at different pH values of the supporting electrolyte should be included.

f. It is necessary to construct a calibration plot.

g. Precision is assessed by repeatability tests (intra and inter-day). Accuracy is evaluated by recovery tests and comparative methods. Review the text.

h. Recovery tests must be performed at more than one concentration level, and these values must be included in the calibration curve.

i. Line 284. How was the LOD value calculated?

j. The results of the recovery tests must be made available in percentage form.

k. The results of the determination on the samples must be presented in a table, together with the recovery tests on the samples.

Author Response

attached file

Round 2

Reviewer 2 Report

 First six(6) sentences should be release and don't revised the first 6 sentences. You must follow a standard abstract.

Real samples analysis must be compared (validated) with commercial instrument such as HPLC. This is a normal procedure for development a new sensor. The instrument is not available in your laboratory is the worst  answer for my question.  

Author Response

attached file

Reviewer 3 Report

Questions were answered, and the manuscript was improved. Therefore, I recommend that it be accepted for publication.

Author Response

Thank you very much for your efforts on the paper